# Tuning the Surface Structure of Polyamide Membranes Using Porous Carbon Nitride Nanoparticles for High-Performance Seawater Desalination

**DOI:** 10.3390/membranes10080163

**Published:** 2020-07-24

**Authors:** Zongyao Zhou, Xiang Li, Digambar B. Shinde, Guan Sheng, Dongwei Lu, Peipei Li, Zhiping Lai

**Affiliations:** Advanced Membranes and Porous Materials Center, Division of Physical Science and Engineering, King Abdullah University of Science and Technology (KAUST), Thuwal 23955-6900, Saudi Arabia; Zongyao.zhou@kaust.edu.sa (Z.Z.); Xiang.li@kaust.edu.sa (X.L.); Digambar.Shinde@kaust.edu.sa (D.B.S.); Guan.sheng@kaust.edu.sa (G.S.); dongwei.lu@kaust.edu.sa (D.L.); peipei.li@kaust.edu.sa (P.L.)

**Keywords:** polyamide membrane, carbon nitride, seawater desalination, mixed matrix membranes, thin film composite

## Abstract

Enhancing the water flux while maintaining the high salt rejection of existing reverse osmosis membranes remains a considerable challenge. Herein, we report the use of a porous carbon nitride (C_3_N_4_) nanoparticle to potentially improve both the water flux and salt rejection of the state-of-the-art polyamide (PA) thin film composite (TFC) membranes. The organic–organic covalent bonds endowed C_3_N_4_ with great compatibility with the PA layer, which positively influenced the customization of interfacial polymerization (IP). Benefitting from the positive effects of C_3_N_4_, a more hydrophilic, more crumpled thin film nanocomposite (TFN) membrane with a larger surface area, and an increased cross-linking degree of PA layer was achieved. Moreover, the uniform porous structure of the C_3_N_4_ embedded in the ”ridge” sections of the PA layer potentially provided additional water channels. All these factors combined provided unprecedented performance for seawater desalination among all the PA-TFC membranes reported thus far. The water permeance of the optimized TFN membrane is 2.1-folds higher than that of the pristine PA-TFC membrane, while the NaCl rejection increased to 99.5% from 98.0%. Our method provided a promising way to improve the performance of the state-of-art PA-TFC membranes in seawater desalination.

## 1. Introduction

Water scarcity is a global issue currently affecting about two-thirds of the world’s population [1]. Over the last half century, reverse osmosis (RO) has been demonstrated as an affordable approach to producing drinking water from seawater [2]. This is mainly owing to the successful development of the start-of-the-art RO membrane: a polyamide thin-film composite (PA-TFC) membrane consisting of a thin and highly cross-linked PA selective layer atop a strong and porous support layer [3,4,5,6,7,8]. However, similar to that of other types of membranes, further improvement in the PA-TFC membrane performance has been limited by the upper limit of the trade-off between permeability versus selectivity [8].

An effective approach to overcoming this limit is based on the concept of mixed matrix membranes. Hoek et al. demonstrated for the first time that by embedding zeolite 4A nanoparticles into the PA layer during interfacial polymerization (IP), the resultant thin film nanocomposite (TFN) membrane exhibited approximately a 50% improvement in water flux without considerable decline in salt rejection [9]. Since then, many other porous materials, such as zeolitic imidazolate frameworks [10,11,12], metal–organic frameworks [13], covalent organic frameworks [14], porous aromatic frameworks [15], and hyper-cross-linked polymers [16], have been extensively studied [17,18,19,20,21]. It is generally believed that these porous materials provide faster transport pathways for water and thus improve the membrane flux. However, there are several challenges in this approach. Firstly, the state-of-the-art PA-TFC membrane is approximately 100 nm thick [2,22]. To synthesize such small nanoparticles is costly and time-consuming. Secondly, it is difficult to increase the particle loading as this may cause particle agglomeration and render the membrane brittle. Lastly, many of these materials are unstable in water, which has raised concerns pertaining to their long-term stability.

Another approach to improving the membrane performance is to optimize the membrane surface structure. The PA layer of a PA-TFC membrane is typically prepared in situ via IP between a diamine and an acyl chloride dissolved in water and an organic solvent, respectively. The organic solvent is immiscible in water. Thus, upon contact, the two species diffuse from each solution to the phase boundary and react there to form a thin film. The reaction is fast, highly exothermic, and typically occurs in an uncontrolled manner; hence, the resultant membrane structure is extremely sensitive to local conditions. One notable feature is the formation of a rough ”ridge-and-valley” surface morphology, which is generally attributed to the inhomogeneous release of reaction heat [22,23,24]. The ridge-and-valley structure is very important for membrane performance because it can increase the membrane surface area [25,26]. Many innovative ideas have been developed to control the membrane surface structure by tuning the process of IP. Zhang et al. reported the formation of Turing patterns to increase the surface area and thus the membrane flux [27]. Jin et al. used a nanostructure-mediated IP process to increase the ridge-and-valley structure [26]. Livingston et al. introduced a sacrificial layer to tune the PA structure [23]. Several other publications have reported the use of a sublayer made of cellulose nanofibers [28], tannic acid/Fe^3+^ nanoscaffold [29], or carbon nanotubes [22,30] to optimize the process of IP. However, although these membranes demonstrate an improved water flux, their NaCl rejection values are generally low. Therefore, these membranes are good for nanofiltration but not suitable for seawater desalination as the latter requires a NaCl rejection value of more than 98.4% to reduce the seawater salinity to a potable level via a one-stage RO process.

Here, we report, for the first time, the use of the highly crystalline carbon nitride (C_3_N_4_) to tune the surface structure of PA membranes to enhance their performance in seawater desalination. As a graphene-analogue material, C_3_N_4_ has a 2D layered structure possessing uniform nanopores with a pore size of 3.11 Å, which is big enough to allow water molecules (kinetic diameter 2.65 Å) to pass through [31]. It has outstanding thermal and chemical stabilities, and is very stable in water and organic solvents [32,33,34]. Its synthesis is easy and inexpensive [35]. C_3_N_4_ has abundant amino ending groups, which render it hydrophilic and chemically very compatible with the PA layer. It was determined that embedding the C_3_N_4_ nanoparticles into the PA layer during the IP process could significantly enlarge the ridge-and-valley surface structure. This effect, as well as the additional transport pathway for water, substantially improved the water permeance of the membrane to 3.2 ± 0.2 Lm^−2^·h^−1^/bar (LMH/bar) and endowed it with an excellent NaCl rejection of 99.5%, which surpassed the performances of most TFN membranes reported to date.

## 2. Materials and Methods

### 2.1. Chemicals

Polysulfone (PSf, Ultrason^®^ S 3010) was purchased from BASF (Frankfurt, Germany). N,N′-dimethylformamide, n-methyl-2-pyrrolidone (NMP), m-phenylenediamine (MPD), trimesoyl chloride (TMC), n-hexane, and melamine were all supplied by Sigma-Aldrich (St. Louis, MO, USA). Non-woven fabric was purchased from Sojitz (Osaka, Japan). NaCl was purchased from VWR Chemicals (Leuven, Belgium). All chemicals were used as received.

### 2.2. Synthesis of C_3_N_4_ Nanoparticles 

C_3_N_4_ was prepared according to the literature [36]. Melamine powder was placed in an alumina crucible and first heated to 400 °C at a heating rate of 4.4 °C/min in an Ar atmosphere. The temperature was maintained at 400 °C for 90 min, and then raised to 600 °C at the same heating rate, held for another 120 min, and then cooled naturally to room temperature. The final product was collected in a yellow powder form (Appendix A).

### 2.3. Preparation of C_3_N_4_ Aqueous Suspension 

A total of 30 mg of C_3_N_4_ powder was dispersed in 20 mL of deionized (DI) water using a probe ultrasonicator operated at 160 W and 25 °C for 5 h. The undispersed C_3_N_4_ powder was removed via centrifugation at 3000 rpm for 15 min. The supernatant solution was sonicated for another 2 h using an ultrasonic cleaner at 70 W and stored at room temperature for subsequent use as a stock solution (Appendix A). The particle concentration in the stock solution was approximately 0.6 mg/mL.

### 2.4. Preparation of PSf Porous Supports 

A PSf ultrafiltration membrane was used as the support layer to prepare the TFC membrane. It was home-made using a previously reported method of non-solvent-induced phase separation [37]. A total of 12 g of PSf was first dried in a vacuum oven overnight at 90 °C and then dissolved in 88 g of NMP to form a homogenous casting solution. The casting solution was placed still for 12 h to remove air bubbles, and then cast on the surface of a non-woven fabric in a closed chamber using a casting knife with a gap of 200 µm. Prior to the casting, the non-woven fabric was wetted with NMP. The relative humidity (RH) and temperature of the casting chamber were maintained at 40% ± 5% RH and 25 ± 5 °C, respectively. The film obtained was immersed immediately into a room-temperature DI water bath and kept there for 12 h to remove most of the solvent. The film was stored at 4 °C for further use. The scanning electron microscopy (SEM) and atomic force microscopy (AFM) images, and the pure water flux of the PSf substrate prepared are shown in the Supporting Information (Appendix A).

### 2.5. IP to Prepare PA-TFC Membrane 

The process of IP was conducted according to our previous protocol [38]. The PSf support was first immersed in an MPD aqueous solution (3.4 wt.%) for 2 min and then withdrawn to naturally drain the MPD solution for 2 min. The excess MPD solution on the support surface was removed using a rubber roller. The TMC solution in hexane (0.15 wt.%) was then poured onto the support surface and reacted for 1 min. Afterwards, the excess TMC solution was removed and the membrane was cured in a hot water bath at 95 °C for 1 min. The final membrane was stored at 4 °C in DI water for further use (Appendix A).

### 2.6. IP to Prepare C_3_N_4_ TFN Membrane 

The C_3_N_4_ stock solution was diluted to concentrations of 0.04 (C_3_N_4_-4), 0.08 (C_3_N_4_-8), and 0.12 (C_3_N_4_-12) mg/mL. Next, MPD was added to the solutions to attain a concentration of 3.4 wt.%. The concentration of the TMC solution was 0.15 wt.%. The remaining steps were identical to those in the preparation of the PA-TFC membrane via IP, as illustrated in Scheme 1.

### 2.7. Characterization 

A high-resolution transmission electron microscopy (HRTEM, FEI Titan, Hillsboro, OR, USA) instrument was operated at 80 kV to obtain the TEM images. The TEM samples were prepared according to our previously reported procedure [22]. The SEM images were obtained using a Magellan XHR instrument (Hillsboro, OR, USA). The SEM samples of the back surface of the PA layer were prepared according to our previous report [22]. All the SEM samples were sputter-coated with 1 nm thick Ir. The dynamic light scattering (DLS) was measured using a Delsa Nano C system (Beckman Coulter, San Diego, CA, USA). The powder X-ray diffraction (PXRD) pattern was collected using a Bruker D8 Advance X-ray powder diffractometer (Karlsruhe, Germany). A Micromeritics ASAP 2420 analyzer (Norcross, GA, USA) was employed to record the N_2_ adsorption isotherms at 77 K. The solid-state nuclear magnetic resonance (NMR) data were obtained using a Bruker Advance 400 WB spectrometer (Karlsruhe, Germany). Atomic force microscopy (Bruker, Dimension Icon, Karlsruhe, Germany) was used to investigate the membrane surface roughness in the tapping mode. X-ray photoelectron spectroscopy (XPS, Amicus Kratos Analytical, ESCA 3400, Kratos, UK) was employed to measure the elemental composition of the PA layer. The water contact angles (WCAs) were measured using the sessile drop method on a drop shape analyzer (Kruss, DSA100, Hamburg, Germany) at room temperature.

### 2.8. Evaluation of RO Desalination Performance 

The water permeance and NaCl rejection of the TFC and TFN membranes were measured using a custom-designed RO permeation cell [10]. After 1.5 h of pre-compaction, the membranes were tested at 15.5 bar and 2000 ppm of NaCl feed solution at 25 °C. The water permeance, *J_w_* (LMH/bar), was calculated using Equation (1).
(1)$$J_{w}=\frac{V}{A\times t\times ∆P}$$
where *V* (L) is the permeate volume collected in a certain period of time, *t* (h), based on the filtration area, *A* (m^2^), and transmembrane pressure drop, Δ*P* (bar). 

The NaCl rejection, *R* (%), was calculated using Equation (2).
(2)$$R=1-\frac{C_{permeate}}{C_{feed}}\times 100\%$$
where $C_{permeate}$ and $C_{feed}$ are the concentrations of the permeate and feed solutions, respectively. 

The salt permeability, *B* (LMH), was calculated using Equation (3).
(3)$$B=J_{w}\times ∆P\times \frac{1-R}{R}$$

All the membranes were tested on at least three samples, and the average values were obtained.

## 3. Results and Discussion

Figure 1A,B show the SEM image and DLS size analysis of the C_3_N_4_ nanoparticles, respectively. The average particle size was around 164 ± 30 nm. The HRTEM image in Figure 1C clearly shows the highly crystalline structure and regular pores, although the selective area electron diffraction displayed in the inset of Figure 1C indicates that the area studied was not a single layer. The high crystallinity was further confirmed by the PXRD pattern in Figure 1D, which contained two characteristic peaks at 13.1 and 27.5°. The PXRD pattern matched well with the reported graphitic C_3_N_4_ pattern [39,40]. Two apparent resonances (δ = 164 ppm and δ = 156 ppm) were detected in the ^13^C cross polarization-magic angle spinning (CP-MAS) spectrum, which were owing to the sp^2^ carbon atoms, as shown in Figure 1E [41]. All these results are in good agreement with the reported results [42], validating that a highly crystalline graphitic C_3_N_4_ was successfully synthesized. The N_2_ physisorption isotherm in Appendix A indicates a very low Brunauer–Emmett–Teller (BET) surface area of 21.5 m^2^/g. This is because the molecular size of N_2_ (3.64 Å) was greater than the pore size of C_3_N_4_. Figure 1F presents the water vapor adsorption isotherm at room temperature and indicates that C_3_N_4_ was relatively hydrophilic. The water uptake at 90% RH was approximately 3 wt.%, which is approximately 15 times higher than that of the reported ZIF-8 nanoparticles [10].

Figure 2 shows the morphologies of the PA-TFC and the C_3_N_4_ TFN membranes at different C_3_N_4_ concentrations, as listed in Table 1. The surface morphologies of all membranes showed the ridge-and-valley feature, but the feature size increased notably with the concentration of C_3_N_4_. The cross-section additionally showed that the height of the PA layer increased with the loading rate of C_3_N_4_. The surface roughness (*Ra*) was measured via AFM. The Ra of the pristine PA-TFC membrane was 20.2 ± 5.6 nm whereas that of C_3_N_4_-12 was 89.7 ± 5.6 nm, representing a considerable increase of four-fold (Table 1). The surface area ratio (SAR), which is defined as the ratio of the total surface area to the projected surface area, increased from 1.4 ± 0.1 for the pristine membrane to 2.3 ± 0.1 for the C_3_N_4_-12 membrane (Table 1). The higher SAR implies a larger specific surface area for transport, which is beneficial to achieving higher water permeation [26].

As shown in Appendix A, the O/N ratio of the membrane surface decreased from 1.46 to 1.31 as the concentration of the C_3_N_4_ nanoparticles increased (Table 1). This can be attributed to two factors: (1) the increase in N resulting from the loading of the nanoparticles into the PA layer; (2) the higher apparent cross-linking degree (CLD) of the PA layer under the effects of the C_3_N_4_ nanoparticles on the process of IP. It is well-known that increasing the CLD will increase the salt rejection [22,43,44]. On the other hand, the uniform nanopores of C_3_N_4_ are also expected to give high salt rejection. Thus, in both cases it is beneficial to the membrane salt rejection, which explains the experimental observation where the NaCl rejection improved as the concentration of C_3_N_4_ nanoparticles increased in the order of pristine TFC < C_3_N_4_-4 < C_3_N_4_-8. Interestingly, the result is different from that reported for TFN membranes with incorporated ZIFs [10,11] or zeolites [45,46]. Those traditional TFN membranes exhibited lower CLDs of the PA layers and lower salt rejection following the addition of the organic/hybrid nanomaterials. Therefore, we tentatively confirmed that, compared with zeolite or other inorganic nanoparticles, the C_3_N_4_ nanoparticles in the present study have better compatibility with the polymer network structure and a much more positive influence on the customization of the IP. Furthermore, the C_3_N_4_ nanoparticles did not compete with MPD or TMC, whereas some nanoparticles modified with active groups [47], interfered with the IP and decreased the CLD of the PA layer, resulting in the loss of salt rejection. The WCA of the PA-TFC membrane was approximately 69 ± 11°. It decreased with the increase in the concentration of C_3_N_4_ (Table 1) and reached 45 ± 9° for the C_3_N_4_-12 membrane. The hydrophilic nature of C_3_N_4_ and the rougher surface should be the two possible reasons for these characteristic results.

To understand the function of C_3_N_4_, the pristine PA-TFC and C_3_N_4_-8 membranes were studied in detail via TEM, as shown in Figure 3. In the PA-TFC membrane, the PA layer exhibited a hollow ridge-and-valley structure with a layer thickness of approximately 20–30 nm (Figure 3A), which was similar to that observed in our previous studies [22] and in the literature [23,48]. In the TFN membrane, the layer thickness was similar, but the ridge-and-valley structure was much larger than that of the PA-TFC membrane. A possible mechanism is proposed as follows. Interestingly, it was found that small C_3_N_4_ particles were clearly embedded into the PA layer, as indicated by the yellow arrows, but the bigger C_3_N_4_ nanoparticles (approximately over 150 nm) always sat in the middle of the ridge-and-valley structure, as shown in Figure 3B1–B4. A possible mechanism is proposed as follows. The large and hydrophilic C_3_N_4_ nanoparticles can trap more MPD solution, as demonstrated by the vapor adsorption isotherm in Figure 1F. It forms a heterogenous reaction zone. The heat generated from the IP reaction further promotes the reaction and breaks the interfacial stability [22,23], thus leading to a larger ridge-and-valley feature. In the case of smaller particles, the influence on the reaction is weak, and thus they are embedded into the PA layer.

The RO desalination performance is shown in Figure 4 and Appendix A. The PA-TFC membrane exhibited a water permeance of 1.7 ± 0.2 LMH/bar with a NaCl rejection of 98.0% ± 0.4%, which were similar to those reported in the literature under the same testing conditions [17,49,50,51,52]. The permeance of the TFN membranes increased with the loading rate of C_3_N_4_. The rejection increased for the C_3_N_4_-4 and C_3_N_4_-8 membranes too but declined for the C_3_N_4_-12 membrane. For the optimal C_3_N_4_-8 membrane, the water permeance reached 3.6 ± 0.2 LMH/bar and the rejection increased to 99.5% ± 0.2%, which is 2.1-folds higher than the permeance of the PA-TFC membrane. Obviously, the enlarged ridge-and-valley structure contributed to the higher water permeance, but its contribution was approximately 1.6-folds according to the SAR. The remaining part was possibly owing to the additional channels provided by the C_3_N_4_ nanoparticles embedded in the PA layer. The higher salt rejections of the C_3_N_4_-4 and C_3_N_4_-8 membranes were because of their higher CLD, as shown in Table 1. However, further increasing the particle loading caused particle agglomeration and defects as shown in Appendix A. Thus, the salt rejection declined. 

The long-term flux of the prepared TFC and TFN membranes was measured to assess their stability during compaction incurred by high-pressure compression. As shown in Appendix A, the normalized steady-state permeance of the membranes approached 0.75–0.80, which is consistent with the literature reported values [49]. There is no significant difference in terms of the steady-state permeance between the pristine TFC membrane and the TFN membranes with various loadings of C_3_N_4_ particles. Thus, the initial reduction in water permeance should be attributed mainly to the compaction of the PSf supports. The C_3_N_4_-8 membrane showed high salt rejection over 99.5% after 48 h testing, which indicated that C_3_N_4_ particles have good compatibility with the PA layer and the resulting TFN membranes have sufficient stability in seawater desalination applications. The performance of the C_3_N_4_-8 membrane was compared with those of other reported TFN systems on the basis of the coefficient (*J_w_*/*B*), as shown in Figure 4B [10,49,50,53,54]. The C_3_N_4_-8 membrane showed an unprecedented *J_w_*/*B* value of up to 15.4, which is superior to that of not only state-of-the art commercial RO membranes, such as Dow^®^ BW30 and Dow^®^ SW30HR, but also other reported TFN membranes. 

## 4. Conclusions

In conclusion, we demonstrated that the highly crystalline C_3_N_4_ could not only provide a faster transport pathway but also effectively tune the structure of PA by tailoring the process of IP to enhance both the water flux and salt rejection. It was found that the crystalline C_3_N_4_ was hydrophilic, containing regular pores that could adsorb a significant amount of water. The organic–organic covalent bonds endowed C_3_N_4_ with great compatibility with the PA layer, which positively influenced the customization of IP. With the increasing loading rate of C_3_N_4_, the size of the “ridge-and-valley” surface structure, the surface area ratio, and the surface hydrophilicity all increased compared to the pristine PA-TFC membrane. Moreover, the cross-linking degree (CLD) also increased with C_3_N_4_ loading rate, which was different from other nanoparticle filler systems. As a result, both the membrane flux and salt rejection were improved. Under the optimal conditions (C_3_N_4_-8), the water permeance was 2.1-folds higher than that of the pristine TFC membrane, while the NaCl rejection increased to 99.5% from 98.0%. Our method thus provided a promising way to improve the performance of the state-of-art PA-TFC membranes in seawater desalination.

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
