# Peer review of "Tuning the Surface Structure of Polyamide Membranes Using Porous Carbon Nitride Nanoparticles for High-Performance Seawater Desalination"

_membranes, 2020, doi:10.3390/membranes10080163_

Round 1

Reviewer 2 Report

The authors employed highly crystalline carbon nitride nanoparticle (NP) to improve the water flux and salt rejection of polyamide (PA) TFC membrane. Embedding the NP into the PA layer enlarges the ridge-and-valley surface structure and thus provides high surface area ratio for additional water transport pathways. It was shown the optimal group is C3N4-8, providing highest salt rejection ratio and second highest water permeability. The authors supplied enough data to find optimal nanoparticle concentration for increased water flux and salt rejection, and explain the phenomena using surface characterization. However, the authors somewhat neglected including the fundamental interpretation about the origin of such improvement. Crosslinked mechanism is somewhat overlooked, and crosslinking ratio calculation based on the XPS could be updated. The authors mentioned about improved mechanical properties, but supporting data need to be included. Therefore, the reviewer recommends the major revision before publication. Other additional comments and questions are shown below.

Other Questions and comments:

  • The authors mention that higher crosslinked density in the PA layer is beneficial for the NaCl rejection. Then why the C3N4-12 group has lower salt rejection than C3N4-8 group (other than particle agglomeration reason)? Is there any chemical restriction or maximum reaction extent?
  • What is the fundamental reason that C3N4 is different from other NP to have higher water permeability without the trade-off of salt rejection ratio?
  • What is the detailed reaction mechanism of crosslinking of C3N4 and PA? Is the proposed one a typical method to use the XPS (the atomic ratio data) to determine crosslinking ratio? How the authors ensure the reduced O/N ratio can be attributed to the crosslinking? Other interpretation includes, increased thickness of C3N4 attaching on the PA layer decrease the O/N ratio?
  • What is your XPS take off angle in this study? Because different XPS take off angle can determine the different thickness of the surface layer and it is important to report which layer you have characterized.
  • The authors mention that the increased particle loading may cause agglomeration and render the membrane brittle in the Introduction. However, according to the TEM image, there do have some particle agglomeration in the studies. Have the authors tested the mechanical strength of the membrane?
  • Is the NP stable in the membrane? Have the authors tested the water permeability and salt rejection in long-term storage in water?
